# A Framework for Assessing Water Security and the Water–Energy–Food Nexus—The Case of Finland

**Mika Marttunen [1,*], Jyri Mustajoki [1] , Suvi Sojamo [2], Lauri Ahopelto [1,2] and Marko Keskinen [2]**

[1] Finnish Environment Institute, Latokartanonkaari 11, FI-00790 Helsinki, Finland; jyri.mustajoki@ymparisto.fi (J.M.); lauri.ahopelto@aalto.fi (L.A.)

[2] Water & Development Research Group, Department of Built Environment, Aalto University, Tietotie 1E, FI-00076 Aalto, Finland; suvi.sojamo@aalto.fi (S.S.); marko.keskinen@aalto.fi (M.K.)

[*] Correspondence: mika.marttunen@ymparisto.fi, Tel.: +358-295-251411

**Abstract:** Water security demands guaranteeing economic, social and environmental sustainability and simultaneously addressing the diversity of risks and threats related to water. Various frameworks have been suggested to support water security assessment. They are typically based on indexes enabling national comparisons; these may, however, oversimplify complex and often contested water issues. We developed a structured and systemic way to assess water security and its future trends via a participatory process. The framework establishes a criteria hierarchy for water security, consisting of four main themes: the state of the water environment; human health and well-being; the sustainability of livelihoods; and the stability, functions and responsibility of society. The framework further enables the analysis of relationships between the water security criteria as well as between water, energy and food security. The framework was applied to a national water security assessment of Finland in 2018 and 2030. Our experience indicates that using the framework collaboratively with stakeholders provides a meaningful way to improve understanding and to facilitate discussion about the state of water security and the actions needed for its improvement.

**Keywords:** water security; water, energy and food nexus; indexes; assessment framework; qualitative assessment

## 1. Introduction

Water security is described as the overall aim for water resources management by a growing number of actors and organisations [1–4]. However, it does not have one all-encompassing definition and its meaning has varied in different contexts and over time. While water security was mostly used to describe specific human security concerns or to set general visions in the 1990s to early 2000s [5], more recently it has been increasingly used to make explicit the actual goals to be achieved with better water resources management [6]. Those goals can be contested and are open to debate between different stakeholders, but water security as a concept has provided a frame for their negotiation [7,8]. Furthermore, it has provided the means to link water to other sectors and their "securities," most importantly to energy security and food security under the concept of the water–energy–food security nexus [9–12].

Today, water security is most commonly understood through four key dimensions, as presented by Hoekstra et al. [6]: economic welfare, social equity, long-term sustainability and water-related risks. Water security thus aims simultaneously to guarantee the three pillars of sustainability (economic, social and environmental) and to address the diversity of risks and threats related to water. This "dual aim" is also recognised by UN-Water, who define water security as "the capacity of a population to safeguard sustainable access to adequate quantities and of acceptable quality water for sustaining

livelihoods, human well-being, and socio-economic development, for ensuring protection against water-borne pollution and water-related disasters, and for preserving ecosystems in a climate of peace and political stability" [3]. Ultimately, water security is thus also linked to the broader foundations of societal security.

Several frameworks have been suggested to support water security assessments and policy making at different levels from local watersheds to national and global scales (e.g., [13–15]). On the one hand, they share similarities with comprehensive nexus [16–18] and water governance assessments (e.g., [19,20]). On the other hand, they build on more quantitative risk and resilience assessment methodologies that utilise water availability, vulnerability and sustainability indexes [7,21,22]. The latter enable illustrative comparisons and trend analyses with comparatively little effort but have been criticised for simplifying complex and often contested water issues and neglecting their social and political aspects (e.g., [7,22–24]). Accordingly, water security as a concept can be both broadened and deepened and its practical applicability improved by combining quantitative and qualitative assessments, considering water security through various alternative aspects and dimensions and engaging different actors and stakeholders, i.e., the end-users in the assessment processes themselves [25–27].

This study builds on these recent recommendations and presents a new assessment framework. The framework is tested in a national water security assessment of Finland. We cover two crucial dimensions to water security that to our knowledge have not been addressed to date. First, even though the importance of considering the linkages between water uses, freshwater ecosystems as well as other sectors is increasingly emphasised (e.g., [9,10,24]), there are currently no studies that assess the internal linkages between the different water security criteria as well as the relations between water, energy and food security. Second, there appears to be no studies combining the systematic assessment of the current and future state of water security taking into account local and global development trends and trajectories.

Existing water security assessment frameworks and indexes have typically focused on such national-level assessments that enable the comparison of water security between different countries and identify issues that require development (e.g., [28,29]). However, the central issues of water security can vary substantially between countries. For example, from policy-making and management perspectives, the common water availability and access indexes and indicators are typically more useful in developing regions that suffer from water scarcity than in water-rich and highly-developed countries. In Finland, a high-quality water infrastructure has been built in recent decades, but its condition is degrading [30]. Therefore, commonly used indicators that measure the coverage of water infrastructure are less useful than an indicator that measures the repair debt of water infrastructure. Thus, our framework aims to capture national and context-specific characteristics.

Water security in Finland is regarded to be at a very high level and Finland is ranked among the top countries in various water security assessments (see, e.g., the water poverty index, [31]; global water security index (GWSI) [28]). In addition, the characteristics of nature and water systems in Finland differ from many other countries. Therefore, urgent management actions can also be very different from other countries. Although the framework is customised to Finnish conditions, it can be modified to be applicable in other countries or regions as well. In addition, the ideas on how to present and visualise the results can be adopted in various types of indicator frameworks.

The paper is structured as follows. In Section 2, we review the existing water security frameworks and indicators and the challenges in their use. Section 3 presents our water security assessment framework and Section 4 the findings from its application in the Finnish water security field. In Section 5, we discuss the strengths, challenges and future development needs related to the application of the framework. Section 6 concludes the paper.

## 2. Water Security Frameworks, Indicators and Challenges

A large number of frameworks and indexes have been developed to assess the sustainability of water management from local to global scale (see the reviews of 95 indicators by Vollmer et al. [21] and 170 indicators by Pires et al. [22]). These indexes enable within-basin comparisons over time or via scenarios and comparisons across basins or countries. Examples of global frameworks are human water security threat maps [32] and global water security indexes [28]. Frameworks that have national or state-wide scope encompass analyses of water security in China [33,34] and Alaska [35]. Vollmer et al. [21,24] developed a freshwater health index and applied it in one Asian river basin, whereas Jensen and Wu [36] developed urban water security indicators (UWSI) and piloted them in Hong Kong and Singapore. In addition to studies with different spatial scales, there are studies that focus on a specific sector, such as agricultural water use [37]. In the majority of earlier water security and sustainability assessments, the target area was located in Africa and Asia, and over the last few years studies have particularly focused on China (e.g., [24,34,36,38]).

The dimensions in the developed frameworks and indexes vary enormously. Chaves and Alipaz [39] proposed the watershed sustainability index (WSI) that incorporates hydrology, environment, life, and policy; each having the parameters of pressure, state, and response. Gain et al. [28] go beyond earlier water scarcity analyses, and use spatial multi-criteria analysis to assess water security in terms of availability, accessibility to services, safety and quality, and management. Sun et al. [33] divide factors affecting sustainable water use into five sub-categories and their key variables: economy, population, water supply and demand (different sub-variables for water supply and demand), land resources, and water pollution and management (quantity of wastewater effluent, sewage treatment capacity, treatment rate of sewage).

Frameworks, indices and indicators can play an important role in various tasks. They can, for example:

- be used as diagnosing tools to identify threats to water security (e.g., [32]),
- be used as management tools giving direction to managerial policy, the allocation of resources and to measure the effectiveness of interventions [31,36,40]
- stimulate policy actions [36],
- improve opportunities for making judgements about the effectiveness of government policy [31]
- provide decision support for better formulation of regional water resources planning [34],
- be powerful tools for stakeholder engagement and communication, and allow policy-makers to communicate policy achievements to the public [36], and
- be important tools for the operationalisation of integrated water resources management [24] and sustainable development goals [41,42].

Despite the fact that many useful indicators and indexes have been developed to assess the sustainability of water management, their use in policy-making is not common [43,44]. There are many challenges in the operationalisation of indicators and indexes that can explain their limited use. Methodological challenges relate to the selection, banding and aggregation of indicators, and consideration of stakeholder participation [38]. The large range of contexts have led to numerous indicators and indexes, which makes the selection of those that are relevant, analytically sound, and measurable difficult [45,46]. Damkjaer and Taylor [47] state that, due to their simplicity, indicators are not meaningful for practical purposes; for instance, intra- and inter-annual variations are typically not considered when measuring water scarcity. Operationalisation can be complicated due to the vague and contested content of the concepts or the overly broad scope (geographical or content) of the analysis [36].

The water, energy and food security nexus (WEF nexus) provides a complementary approach to water security and its linkages to energy and food (e.g., [9–11,48]). The nexus approach can be considered both an analytical framework and a governance approach, with the latter promoting policy coherence and collaboration between different sectors [11]. When used as an analytical framework,

the nexus aims to identify synergies and trade-offs within the subsystems that constitute the overall water–energy–food security nexus system (e.g., [10,49–51]). Many literature reviews have been realised during the last few years covering concepts and methodologies related to the WEF nexus (e.g., [11,51–55]). It is interesting to note that according to these reviews, the use of WEF nexus methods to systematically evaluate water, energy and food interlinkages has been limited [53], and an empirical WEF nexus research has not yet validated claims that nexus approaches can improve resource management and governance outcomes [54].

## 3. Assessment Framework

The process of applying our proposed framework is presented in Figure 1. The first phase (framing) includes the identification of the problem and its elements (step 1 in Figure 1), and defining the assessment criteria and structuring them into a hierarchical form (step 3). This phase also includes the identification of the stakeholders (step 2) whose involvement in the development process is considered important, and the determination of end-users of the information produced by the assessment. The second phase is the actual assessment, where each criterion is assessed in terms of various dimensions (steps 4–7). In the last phase, analysis, the status of the water security is obtained on the basis of the overall view of the assessment (step 8) and future conclusions are made (step 9).

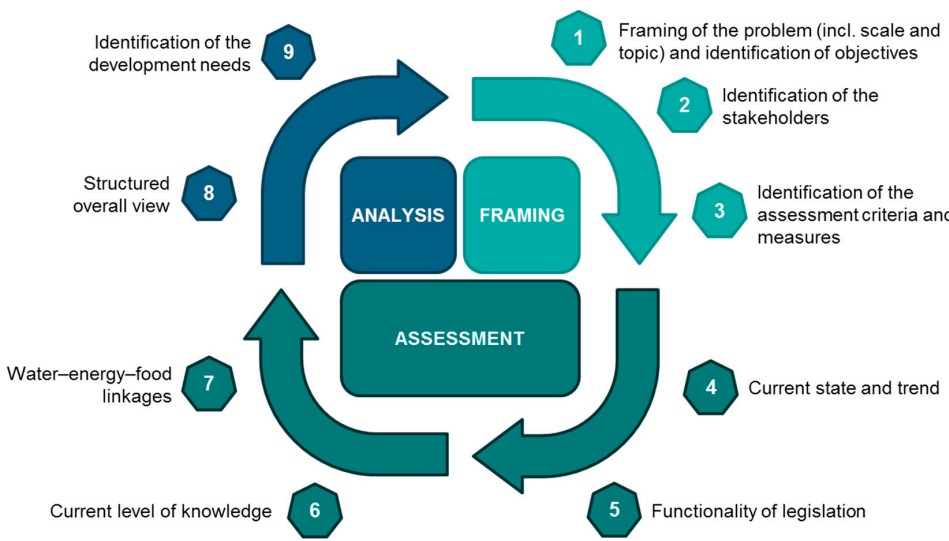

**Figure 1.** The steps of the water security assessment process.

Development of the framework was a combination of extensive research by the authors and other researchers in the Winland research project (2016–2019), looking at future energy, food and water security in Finland, and consultation and co-creation with key stakeholders. The research included, for example, analysis of strategy documents from different ministries (Ministry of Environment, Ministry of Agriculture and Forestry, Ministry of Foreign Affairs), a literature review of 32 international papers dealing with indicators, frameworks, metrics and indexes for measuring water security and sustainable water use, and the identification of 25 drivers that may affect water security of Finland (12 of them were selected for the analysis).

The research findings were discussed with stakeholders at regular intervals in workshops and interviews. Altogether four stakeholder workshops were organised, of which the first two dealt with identification of drivers and assessment criteria for water security and development of a criteria hierarchy for water security (explained in Section 3.1). The hierarchy was further developed in the third workshop, before which the participants, who came from ministries, research institutes and national security organisations, were also able to comment on the content and structure of the hierarchy in a web survey (e.g., whether there is a need to add or remove criteria or to change their names).

The hierarchy was modified on the basis of the comments received. In order to get feedback from the potential end-users on the assessment framework and its application opportunities, we carried out six high-level interviews: four directors from the Finnish Environment Institute (SYKE) and two officials with lengthy experience in national and international water management tasks from ministries. After the interviews, a fourth workshop focusing on the actual assessment of Finnish water security was arranged, and the participants were still able to comment on the hierarchy in the workshop after which the framework was finalised.

We also developed an Excel tool to support the operationalisation of the framework. The tool consists of different sheets, including a sheet for defining the criteria structure and separate sheets for each of the assessment dimensions. In addition, there is another separate sheet for summarising the results, and various others for visualising the results of the analysis, for example, with the possibility to create an individual assessment card for each criterion (see Appendix A, Figure A1).

### 3.1. The Criteria Hierarchy for the Assessment

Water security is a broad concept covering all water-related sectors, and taking all the different issues into account separately would make the analysis too complex. Our main aim in the development of the criteria hierarchy was to find a balance between the comprehensiveness and practicality of the analysis. Therefore, we decided to use an approach where most criteria were defined so that they consisted of several issues. For instance, sustainable use of natural resources includes mining, forestry, fishery and peat extraction. The development process was iterative, and modifications to the hierarchy were made after the workshops and the interviews. Additions to the hierarchy typically also required some restructuring or rewording of the existing criteria.

As water security has differing meanings for different actors, it is critically important to agree on its key criteria at first. The final criteria hierarchy used in this study consists of 18 criteria classified under four main criteria: 1) State of the water environment; 2) Human health and well-being; 3) Sustainability of livelihoods; and 4) Stability, functions and responsibility of society (Figure 2, see descriptions of the criteria in Table A1, Appendix B). The assessment was demarcated to cover the Finnish water security field.

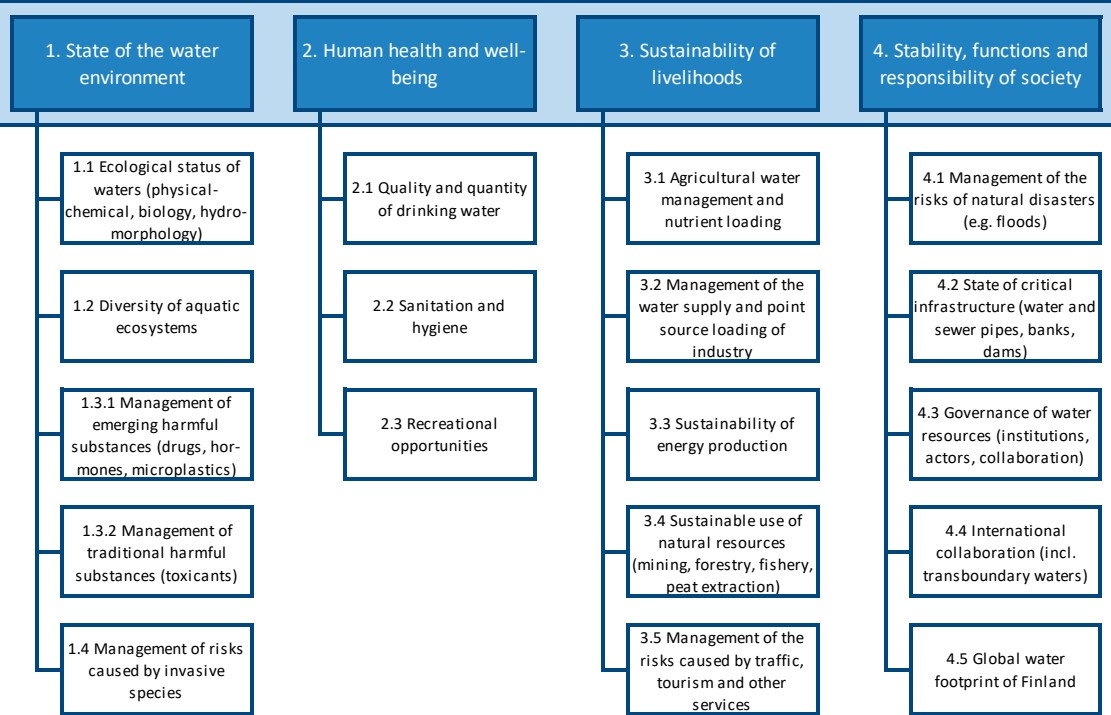

**Figure 2.** The criteria hierarchy used in the water security framework.

As mentioned, the hierarchy evolved somewhat during the process. For example, at first the framing seemed to be unambiguous, but there were still lively discussions on how to deal with the impacts related to global water security. The harmful substances were initially considered as one criterion, but this was divided into two parts as its assessment as one entity was difficult; traditional harmful substances (e.g., heavy metals) are a decreasing environmental problem in Finland, whereas new harmful substances (e.g., drugs, hormones, microplastics) are an emerging problem. At first the management of the environmental risks of traffic (e.g., oil and chemical accidents in roads, railways and waterways) was only implicitly considered under the other criteria, but it was finally included as an explicit element of the framework, also including tourism and other services.

### 3.2. Assessment Dimensions

The aim of the assessment is to identify the issues that are currently managed (reasonably) well, as well as the issues that have to be improved to achieve good status or to prevent the deterioration of their current status in the near future. Thus, we did not calculate an aggregated assessment index for overall water security like Gain et al. [28], but instead assessed each water security criterion separately.

The process of setting suitable dimensions for assessing each criterion was also iterative, in a similar way to the development of the criteria hierarchy. The challenge here was also to get a set of meaningful and relevant assessment dimensions that are able to capture central aspects related to water security. The current state of the criterion and its future trend were selected as natural starting points for the assessment. The current level of knowledge and functionality of legislation (i.e., how up-to-date the legislation is and how flexible it is to changing circumstances) were included in the set of assessment dimensions, as a lack of knowledge and outdated legislation are both clear signals that further actions are needed. The linkages between the criteria as well as the linkages to energy and food security were also included, as one of our objectives was to improve the understanding of the complex relationships between the water, energy and food security nexus.

Figure 3 provides a summary of the assessment dimensions and outcomes of the assessment. The dimensions and especially their assessment scales are partly case-dependent; therefore, they are explained in more detail in Section 4 along with the Finnish water security case.

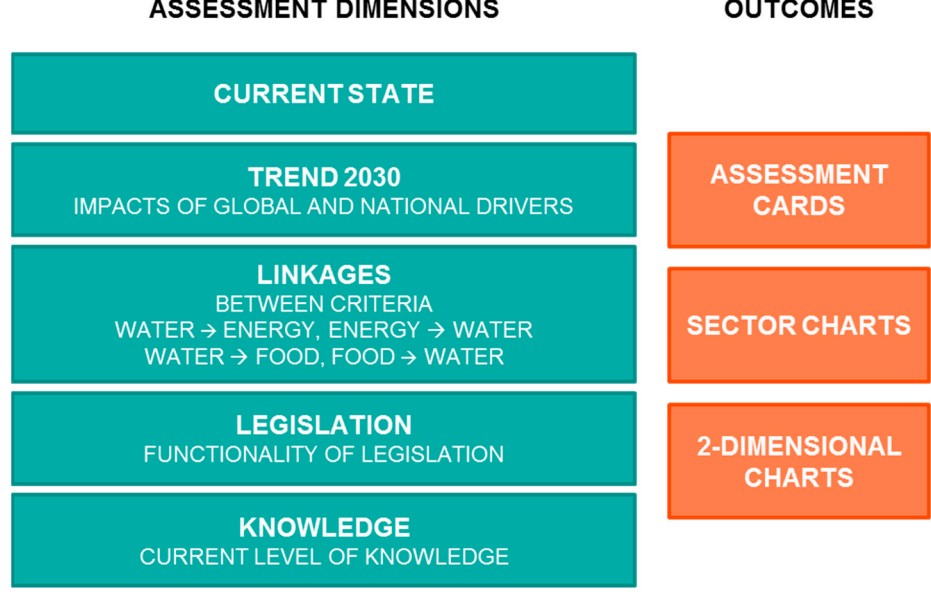

**Figure 3.** Summary of the assessment dimensions and outcomes of the assessment.

The assessment was based on expert judgments informed by various indicators and research reports. We did not, however, create any numerical scales for the indicators to describe their current

state. Instead, we used a five-point generic Likert scale complemented with qualitative description for the assessment. The reason for this was that we wanted to make explicit the justification for the state estimates of each criterion. For this purpose, single indicators would have narrowed the analysis too much (e.g., [47]). Furthermore, the use of numerical scales would also have required an expert assessment of its endpoints, and their choice would also have been partly subjective.

### 3.3. Presenting the Results

A visual, illustrative presentation of the results is one of the key factors when discussing the assessment results with stakeholders and decision-makers, and it was therefore one of the main reasons for developing the Excel tool for the assessment. Appendix A provides examples of various visualisations in the Excel tool. For the overall comparison of different criteria, the tool provides a summary table, which includes results from all assessments in a plus/minus form. The matrix is also coloured according to the assessment scales, which makes it easy to see which criteria perform the best and which are the worst in each of the dimensions. The tool presents matrix information as a collection of sector graphs for each criterion. This kind of visualisation makes it easier to grasp the overall performance of each criterion at a glance, as all the assessments of each criterion are collected in a compact circle instead of showing them as a single line in the matrix.

For analysing a single criterion in a more detailed way, the tool automatically creates an assessment card (i.e., a score card) for each criterion (Appendix A, Figure A1). In that card, all the information regarding a single criterion is offered in a structured form. The card also includes qualitative descriptions of the assessments, which are omitted from the summary tables. At the end of the card, there is a summary of the information and the sector presentation of the assessment values.

## 4. Assessing Finnish Water Security with the Framework

### 4.1. Application Process

Utilising the available indicator data (see Appendix B, Table A2), related assessments and data collected in the first three workshops, preliminary estimates for the criteria on the various assessment dimensions were first made by the authors. These estimates were then presented to experts from different sectors. For instance, estimates of the natural disaster management were commented on by a person who works on flood risk management issues. The preliminary estimates were discussed in four interviews with water management experts at SYKE and two in the associated ministries (see also Section 3).

### 4.2. Assessment of the Current State and Trend, Level of Knowledge and Functionality of Legislation

The results regarding the state, legislation and knowledge of water security criteria are presented in Table 1. Besides the estimates presented in Table 1, the qualitative explanation behind each estimate was included to elaborate on the reasoning behind the estimate. Our analysis highlighted that there are several water security issues in Finland that need more attention. For example, the loading of nutrients and solid substances from agriculture and the use of natural resources (forestry, peat extraction) are currently at quite high levels, and actions are needed to improve the situation, especially if the targets of the EU Water Framework Directive (WFD) [56] are to be reached. The quality of the drinking water is currently very good, but the constantly increasing repair debt of the water infrastructure is expected to have a negative effect on the situation in the future (e.g., [30]). There are also various harmful substances (drugs, hormones, microplastics) that are an emerging problem (e.g., [57]). In addition, there is not much knowledge about their impact chains and long-term effects, and therefore this is a topic that needs further research.

**Table 1.** Assessment of the criteria for Finnish water security.

| Water Security Assessment | State | | Legislation | Knowledge |
|---|---|---|---|---|
| | Current State | Trend 2030 (State Change) | Functiona- Lity of Legislation | State of Knowledge |
| **1. State of the water environment** | | | | |
| 1.1 Ecological status of water (physical-chemical, biology, hydro-morphology) | - | - | - | - |
| 1.2 Diversity of aquatic ecosystems | - - | - | - | + |
| 1.3.1 Management of emerging harmful substances (drugs, hormones, microplastics) | - - | - | - | - - |
| 1.3.2 Management of traditional harmful substances (toxicants) | - | - | o | + |
| 1.4 Management of risks caused by invasive species | - | - | - | - |
| **2. Human health and well-being** | | | | |
| 2.1 Quality and quantity of drinking water | + | - - | o | o |
| 2.2 Sanitation and hygiene | + | - | o | o |
| 2.3 Recreational opportunities | + | - | o | - |
| **3. Sustainability of livelihoods** | | | | |
| 3.1 Agricultural water management and nutrient loading | - - | - | - | - |
| 3.2 Management of the water supply and point source loading of industry | - | - | - | + |
| 3.3 Sustainability of energy production | - | o | - | o |
| 3.4 Sustainable use of natural resources (mining, forestry, fishery, peat extraction) | - - | - | - | + |
| 3.5 Management of the risks caused by traffic, tourism and other services | o | - | + | - |
| **4. Stability, functions and responsibility of society** | | | | |
| 4.1 Management of the risks of natural disasters (e.g., floods) | o | - | o | o |
| 4.2 State of the critical infrastructure (water and sewer pipes, banks, dams) | o | - | - | - |
| 4.3 Governance of water resources (institutions, actors, collaboration) | o | - | o | o |
| 4.4 International collaboration (including transboundary waters) | + | + | o | o |
| 4.5 Global water footprint of Finland | - | + | o | o |

**Scales for the assessment dimensions**

**Current state**

| | |
|---|---|
| + + | Current state excellent or exceeds the target level |
| + | Current state predominantly good or at the target level |
| o | Current state is ok or close to the target level |
| - | Current state is satisfactory or worse than the target level |
| - - | Current state is weak or considerably below the target level |

**Trend 2030 (State change)**

| | |
|---|---|
| + + | State is expected to improve significantly by 2030 |
| + | State is expected to improve somewhat by 2030 |
| o | State is expected to remain same as now in 2030 |
| - | State is expected to weaken somewhat by 2030 |
| - - | State is expected to weaken significantly by 2030 |

**Functionality of legislation**

| | |
|---|---|
| + + | Legislation works well, is flexible and makes it possible to make justified decisions also in changing conditions |
| + | Legislation works well in current conditions |
| o | Legislation works quite well, but needs some updating |
| - | Legislation is partly outdated and needs updating |
| - - | Legislation is outdated and greatly needs updating |

**State of knowledge**

| | |
|---|---|
| + + | The level of understanding is very good, enabling the choice and implementation of the cost-effective measures |
| + | The level of understanding is good, and there is little need for additional research |
| o | The level of understanding is moderate, but new research can help to identify cost-efficient measures |
| - | The level of understanding is quite poor and more research is needed to understand the system and find cost-effective measures |
| - - | The level of understanding is poor and much more research is needed to understand the system and find cost-effective measures |

In the assessment of the current state, the main challenge was to decide on the good or acceptable state (baseline) of each criterion. With some criteria, generally approved goals or measures exist that were used in the assessment. For example, the WFD provides a classification system according to which the ecological state of the water can be assessed. To support the assessment, we collected a list of existing indicators related to each criterion (Appendix B, Table A2). However, with most of the criteria, the assessment had to be made qualitatively as an expert judgement. The assessment of the current state was made using the five-point Likert scale, with a range from "*Current state is weak or considerably*

*below the target level*" to "*Current state is excellent or exceeds the target level*". With regard to some criteria, a precise target level exists. For the ecological status of water, the target level is good ecological status according to the Water Framework Directive. For some other criteria (e.g., sustainability of energy production), target levels were more vague and the assessment was based on experts' views about the prevailing situation compared to an ideal situation. The baseline level of the assessment is "*The state is OK or close to the target level*".

It was decided that the time span for the trend analysis should run until 2030. While this is the target year for UN SDGs, it is only a tentative date, which is far enough in the future so that the trends that can be observed today might have more prominent impacts (positive or negative) on the state of the criteria. However, one should note that the trend up to 2030 is just an estimate and should not be considered literally as a representation of the future. For the systematic assessment of the trends, we created a list of global and national drivers that have previously been identified that will potentially affect water security in the near future. The global drivers included in the framework were:

- Climate change
- Population growth
- Globalisation (transfer of people and goods)
- Digitalisation
- Development of science and technology
- Increase in the use of harmful substances and chemicals

Similarly, the nationally important drivers or trends included into the framework were:

- Urbanisation
- Ageing of population
- Low investment rate to the renewal of water infrastructure
- Deterioration of the drainage systems of agricultural fields
- Intensification and centralisation of agriculture
- Diminishing water expertise in environmental administration

Each driver's impact on each criterion was assessed using a five-point scale ranging from "*Significant negative impact*" to "*Significant positive impact*" (i.e., a noticeably or measurably large amount). After the driver/trend-specific assessment, the overall impact of the different types of drivers on each criterion was assessed holistically using a five-point scale ranging from "*State is expected to weaken significantly by 2030*" to "*State is expected to improve significantly by 2030*".

Five-point scales were used for current level of knowledge and functionality of legislation. In terms of the level of knowledge, the scale was based on how much knowledge there is for making informed decisions regarding the necessary measures, and how much further research is needed. The end points of the scale are "*The level of understanding is poor and much more research is needed to understand the system and to find cost-effective measures*" and "*The level of understanding is very good, enabling the choice and implementation of the cost-effective measures*". For the assessment of the functionality of legislation, the focus was on how up-to-date the existing legislation is, and whether there is a need to renew it. The end points of the scale are "*Legislation is outdated and there is a great need to update it*" and "*Legislation works well, is flexible and makes it possible to make justified decisions in changing conditions*".

The tool provides the possibility to create two-dimensional tables, in which the performances of the criteria can be classified simultaneously according to any two assessment dimensions. One can, for example, set the current state to the x-axis and the trend to the y-axis (Figure 4). Then, the most critical criteria requiring urgent actions are those in the lower-left corner, where the current state is poor ("*Current state is poor or considerably worse than the target level*") and the trend is alarming ("*State is expected to weaken significantly by 2030*"). Those criteria where the current state is good but the trend is alarming (lower-right corner) might require special attention, as the good current state may give a false feeling of satisfaction that things will be OK in the future, even if this is not necessarily the case.

| Trend 2030 (state change) | -- | - | 0 | + | ++ |
|---|---|---|---|---|---|
| ++ | | | | | |
| + | | - 4.5 Global water footprint of Finland | | - 4.4 International collaboration | |
| 0 | | - 3.3 Sustainability of energy production | | | |
| - | - 1.2 Diversity of aquatic ecosystems<br>- 1.3.1 Management of emerging harmful substances<br>- 3.1 Agricultural water management and nutrient loading<br>- 3.4 Sustainable use of natural resources | - 1.1 Ecological status of waters<br>- 1.3.2 Management of traditional harmful substances<br>- 1.4 Management of risks caused by invasive species<br>- 3.2 Management of the water supply and point source loading of industry | - 3.5 Management of the risks caused by services<br>- 4.1 Management of the risks of natural disasters<br>- 4.2 State of critical infrastructure<br>- 4.3 Governance of water resources | - 2.2 Sanitation and hygiene<br>- 2.3 Local recreational opportunities | |
| -- | | | | - 2.1 Quality and quantity of drinking water | |
| | **--** | **-** | **0** | **+** | **++** |
| | | | **Current state** | | |

**Figure 4.** A two-dimensional chart for identifying the criteria that need the most attention. Note that the width and height of each column is adapted to the amount of text.

## 4.3. Linkages between the Water Security Criteria

To support the identification of the linkages between the water security criteria, the Excel tool provides a cross tabulation matrix of them (Figure 5). The aim is to point out that the systemic nature of water security also requires the analysis of interrelationships between the criteria. In practice, the impact can either be positive or negative. A positive relationship means that the positive movement of a criterion in a row leads to a positive movement of a criterion in a column, and, correspondingly, a negative movement in a row leads to a negative movement in a column. In a negative relationship, the impacts are the other way around. For example, improvement in the ecological status of water improves the diversity of aquatic ecosystems as the living conditions for salmonids that prefer oligotrophic conditions improve.

When considering the internal linkages between the criteria, ecological status, diversity of aquatic ecosystems, quality and quantity of drinking water and recreational opportunities are the criteria that are most affected by the other criteria. The governance of water resources has the widest range of positive relationships as it affects all other water security criteria. International collaboration, such as within the EU, can also affect water security in several ways: (i) it can lead to new directives to protect water or limit emissions, (ii) it can lead to joint projects that aim to improve the status of water (e.g., the Baltic Sea), and (iii) research funding from the EU plays an important role in many restoration and rehabilitation projects. The possible deterioration of ecological status may either restrict human activities and industry or cause stricter emission permits. Flood risk management includes measures such as dredging, building flood banks and regulating lakes and rivers, which can have negative environmental impacts.

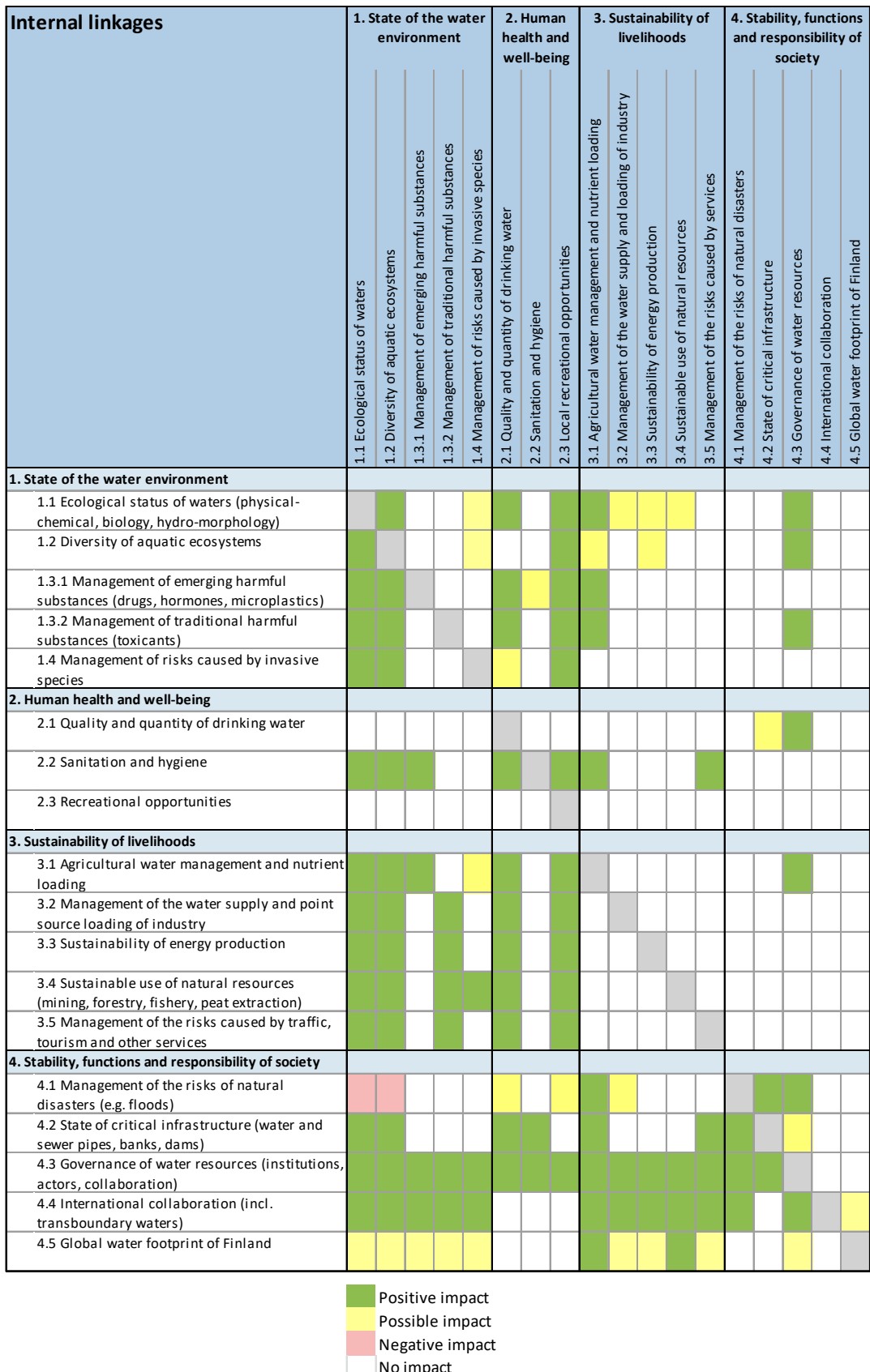

**Figure 5.** Internal linkages between the 18 assessment criteria.

### 4.4. Linkages between Water Security Criteria and Food and Energy Security

The assessment of linkages in the water–energy–food security nexus was bi-directional so that for each water security criterion, we identified whether it has an impact on food/energy security, and also whether food/energy security has an impact on this particular water security criterion. In practice, the impact can either be positive or negative, but we did not take a stance on the direction of the linkage. The reason was that in many cases there can be linkages in both directions, as energy and food security are such broad areas. For example, an improved state of water can improve food security through improved water quality, but on the other hand this can also reduce food security through actions that are required for farming to improve water quality. In our case, it was most important to identify that this link exists, but its direction was not that important, as both directions increase the significance of the criterion. Thus, in the assessment of linkages, we applied scale "*No or weak linkage*", "*Positive or negative linkage*", or "*Significant positive or negative linkage*".

It is noteworthy that we analysed the linkages between each water security criterion and energy security and food security systematically. For example, for energy security, all types of energy sources were covered (e.g., hydro power, nuclear power, wind power, coal, natural gas, peat), which, of course, complicated the assessment as there were several issues that had to be borne in mind simultaneously, and because the linkages to different energy sources are not identical. In terms of the linkages between food/energy security and the rather specific criteria of water security in particular, it was sometimes hard to concretise the actual impact, which further supported the use of a non-directional scale. Therefore, we instructed experts to narrow down the perspective and, for example, with regard to the energy-water nexus, consider only the amount of energy production instead of energy security in general, which is a broad concept.

The linkages between water and energy security (Table 2) stem mainly from the conflict of interest between hydropower production and the ecological criteria in harnessed and regulated watercourses. On the other hand, the watercourse regulation developed for hydropower production can play an important role, for example, in flood prevention. However, all these issues have been under debate in Finland for several decades and therefore have a central role in the governance of water resources. In terms of the linkages between water and food security, the obvious ones are those between farming and the status of the water (ecological status and biodiversity). This link can be noted both in the livelihood criterion and the environmental criterion. This issue is also related to the governance of water resources.

**Table 2.** Linkages between water and food/energy security.

| Water Security Assessment | Linkages with Energy | | Linkages with Food | |
|---|---|---|---|---|
| | Water→ Energy | Energy→ Water | Water→ Food | Food→ Water |
| **1. State of the water environment** | | | | |
| 1.1 Ecological status of water (physical-chemical, biology, hydro-morphology) | * * | * * | * | * * |
| 1.2 Diversity of aquatic ecosystems | * * | * * | o | * * |
| 1.3.1 Management of emerging harmful substances (drugs, hormones, microplastics) | o | o | * | * |
| 1.3.2 Management of traditional harmful substances (toxicants) | o | * | * | * |
| 1.4 Management of risks caused by invasive species | o | * | * | o |
| **2. Human health and well-being** | | | | |
| 2.1 Quality and quantity of drinking water | o | o | * | * |
| 2.2 Sanitation and hygiene | o | o | * | o |
| 2.3 Recreational opportunities | * | * | o | * * |
| **3. Sustainability of livelihoods** | | | | |
| 3.1 Agricultural water management (water supply, consumption, drainage) and nutrient loading | o | * | * * | * * |
| 3.2 Management of the water supply and point source loading of industry | o | o | * | * |
| 3.3 Sustainability of energy production | * * | * * | * | * |
| 3.4 Sustainable use of natural resources (mining, forestry, fishery, peat extraction) | * | * * | * | o |
| 3.5 Management of the risks caused by traffic, tourism and other services | o | * | o | * |

**Table 2.** *Cont.*

| Water Security Assessment | Linkages with Energy | | Linkages with Food | |
|---|---|---|---|---|
| | Water→ Energy | Energy→ Water | Water→ Food | Food→ Water |
| **4. Stability, functions and responsibility of society** | | | | |
| 4.1 Management of the risks of natural disasters (e.g., floods) | * | * * | o | o |
| 4.2 State of critical infrastructure (water and sewer pipes, banks, dams) | o | * * | * | o |
| 4.3 Governance of water resources (institutions, actors, collaboration) | * | * * | * | * * |
| 4.4 International collaboration (including transboundary waters) | * * | * * | o | o |
| 4.5 Global water footprint of Finland | * | * | * | o |
| **Scale** | | | | |
| * * | Significant positive or negative linkage | | | |
| * | Positive or negative linkage | | | |
| o | No or weak linkage | | | |

## 5. Discussion

### 5.1. Methodological Discussion: Pros and Cons of the Assessment Framework

The application of our framework can support water security assessment in several ways. It helps to concretise the abstract water security concept, and provides a systematic and visual way to discuss different aspects of water security. Use of the broad definition of water security also enables us to cover issues that are less frequently examined, such as stability, functions and responsibility of society, and functionality of legislation. A comprehensive understanding of water-related issues and their relative importance is vital for rational decision-making. To support this, the tool combines the knowledge of experts from various fields into a coherent overall picture. In public debate, less important issues sometimes get lots of attention, whereas much more important issues are ignored. In this respect, the framework assists in highlighting issues that should be focused on. The framework also includes a way to link water security with food and energy security, thus promoting the nexus approach. A systemic and comprehensive approach can be useful in foresight and risk management processes by providing a structuring framework for discussions and assessment.

This was the first time a water security assessment framework was developed and an assessment conducted in Finland. Therefore, it is not surprising that deciding on the water security criteria, assessment dimensions and the scales of the criteria was a rather difficult and iterative process. A central question was the trade-off between the compactness and the level of detail of the analysis. In the final framework, the criteria are large entities (e.g., different types of industries/use of natural resources have been combined) to facilitate the implementation of the analysis and to reduce the workload. This had a side effect, as setting an overall estimate for the combined criteria was sometimes difficult because various sectors may have advanced differently or may have diverging impacts regarding a water security criterion. One challenge was to ensure that the estimates assigned by several experts were coherent with each other. Different experts may interpret the qualitative scales in different ways, which may undermine the comparability of the results of the various criteria. To avoid that, we recommend that experts are interviewed personally, as it allows for a reduction of the risk of misunderstandings and provides the opportunity to ask about further arguments. Furthermore, the interpretation of the results was challenging due to the general nature of some criteria. However, as our aim was to get a picture of the overall water security situation in Finland, a certain generality in the assessment was considered acceptable. In the forthcoming cases, it is important to judge on a case-by-case basis if there is a need to split a combined criterion into sub-criteria.

Compared to many other recently developed water security frameworks (e.g., [28,34,35]), our approach is more comprehensive (including, e.g., the water–energy–food nexus and governance), more visual, and involves stakeholders more intensively. It is also more qualitative and subjective (expert judgements are used for example in the overall assessment of the state of criteria consisting of several issues), which also enables the inclusion of non-measurable issues in the assessment. On the

other hand, this brings a certain kind of vagueness to the estimates and thus may reduce the credibility of the assessment from the perspective of external actors. Due to subjectivity, it is likely that various experts may have differing opinions about the issues. However, one aim of the framework is to identify the issues of disagreement, for which the subjective assessment can act as a catalyst for discussion.

There are examples of using multi-criteria decision analysis (MCDA) to aggregate various water security criteria (e.g., [28,58]). We also discussed whether the application of MCDA would be useful in our case, but decided not to aggregate the criteria. First, our analysis is semi-quantitative and thus calculating a numerical index would have given a false impression of the accuracy of the analysis. Second, there is an obvious risk of double counting as there are several criteria that are closely linked to each other. For instance, all categories of DPSIR diagram (Drivers, Pressures, States, Impacts, Responses) are covered by the criteria of the framework. Third, we considered it easier for stakeholders to follow our analysis if we draw conclusions from the criterion-based analysis rather than from a more aggregated analysis.

## 5.2. Discussion of Water Security in Finland

Finland has been regarded as a model country for water supply. Currently, about 90% of Finnish households are connected to centralised water supply networks, about 85% of the inhabitants are linked to the sewerage and centralised wastewater treatment, and the rest have decentralised and personal systems [30,59]. However, the infrastructure is ageing and there are estimates that the current investments for water supply repair and replacement are 0.5–1% of the capital value of the networks (120 million euros) when the required level should be at least 2–3% (320 million euros). Ageing or poorly maintained distribution systems can cause the quality of piped drinking water to deteriorate below acceptable levels and cause serious health risks. The state of water supply networks affects the state assessment of two water security criteria (2.1 Quality and quantity of drinking water and 4.2 State of critical infrastructure) and is an issue that should be at the heart of water management and resource allocation in the near future.

Another infrastructure-related issue of central importance is Agricultural water management and nutrient loading (Criterion 3.1). As the role of state in co-funding main drainage investments has decreased and the profitability of agriculture has been poor for a long time, the investments in agricultural drainage infrastructure have not been at an adequate level. Inadequate drainage can reduce crops, make it more difficult for machinery to move in the fields and increase nutrient loads. There is also a lack of knowledge about the state of the main drainage systems as the last national investigation was performed at the beginning of 1990 [60]; therefore, this topic should be a research priority.

Sustainable use of natural resources is a widely accepted goal that applies to different ministries in Finland. However, whether economic, social and environmental objectives have been balanced well enough has been a topic of continuous debate. As forestry is one of the pillars of the economy in Finland, intensive forest management, including the ditching of peatlands, is practiced throughout the country, except in northernmost Lapland. Forest felling is also expected to increase to meet the envisioned national bioeconomy targets, which is causing growing conflict between economic and environmental objectives [61]. There is also a widespread concern about the increased activities of international mining companies. There have been shortcomings in the assessment and management of the impacts of mining water, both on the part of authorities and the mining operators. The responsibilities and guarantees of mining companies regarding incident terms of prevention and preparedness, remediation and post-closure care remain inadequate [62].

Climate change is the most dominant driver, as it has negative impacts on several water security criteria, most notably on the state of the water environment. Climate change will directly affect lake ecosystems through higher temperatures and changes in the hydrological cycle [63]. As a result, nutrient and organic matter runoff to water courses is expected to increase, which in part increases the risk of eutrophication. In addition, higher summer temperatures may cause changes in the flora and

fauna in lakes and rivers. The distribution of salmonid fish species, which favour cool water, may move north, for example. Climate change also increases the potential for harmful invasive species to spread.

The analysis of legislation revealed that, out of the 18 water security criteria assessed for legislation, nine of them were partially outdated (Table 1), so the need to update the regulations is moderately high. Regarding the nexuses, four water security criteria were estimated to be significantly linked to energy production. For example, improving the Diversity of Aquatic ecosystem requires an increase in environmental flows in many rivers, which cause losses to hydro power production. On the other hand, energy production was estimated to have a significant link to eight water security criteria.

The assessment highlights the development needs for policy-makers by providing a structured and detailed understanding of the state of water security in Finland.

*5.3. Limitations and Ways Forward*

We emphasise that the aim of our analysis is to provide one viewpoint for the Finnish water security discussion, and that there is uncertainty and subjectivity in the estimates. They are not designed to be an "ultimate truth". Thus, the estimates can and should be contested. Actually, an important objective of the framework is to identify the issues in which there are disagreements between stakeholders. Thus, pinpointing the estimates that are not in line with someone's personal view is one way to promote and direct discussion regarding water security. The assessment conducted with the Excel tool is available for feedback on the internet, and suggestions for adjusting the assessments are welcome.

Our nexus analysis was qualitative and was done at a very general national level, and should therefore be seen as indicative only. We assume that realising the analysis in a much more limited geographical area, e.g., a watershed, could produce more concrete and more applicable results. Further practical testing of the approach would also provide additional information about which water security criteria and dimensions the assessment should address, as all the dimensions of the framework may not be relevant in every case.

It should be noted that the assessment is scaled to Finnish conditions, with the aim of identifying critical issues and development needs. On a global scale, water-related challenges in Finland are relatively minor compared to those in many other countries. For instance, Finland has been ranked one of the top countries in the world with regard to water security several times (e.g., [28,31]). Therefore, if the ranges of the scales were set up to cover all the countries in the world, they would not have been discriminating enough for our purposes.

Defining a more comprehensive set of indicators would make the assessment more objective and transparent. The challenge is that a large number of metrics are needed, as metrics typically describe only a small part of the content of one criterion.

If the development of the framework was laborious and demanding, its use may be too. As the framework aims to be comprehensive, it covers a wide variety of environmental, socioeconomic and institutional criteria. There are very few experts who have expertise in all of these criteria, and therefore the application of the framework should be realised in a group consisting of representatives from different fields. Thus, the analysis could be updated from time to time to find out whether the results are still valid and how the state of the various criteria has developed, as well as whether any new issues or threats have emerged that should be given particular attention in water policy-making and management.

## 6. Conclusions

In this paper, we have presented a comprehensive framework and an Excel tool to assess water security in Finland. We have described the elements of the framework and the main results of analysis utilising it, as well as the challenges we faced during the development process.

Compared to the majority of earlier frameworks developed for water security or sustainability assessments, ours is more comprehensive, less quantitative, more visual and more subjective. The framework helps us to compare different dimensions of water security, facilitate discussion

among stakeholders, and identify central research and development needs through a process that is less biased. The framework developed responds to the need for systemic approaches in foresight and risk management.

Applying the framework in a collaborative way with different stakeholders provides a structured yet laborious way to increase managers' and policy-makers' understanding about the different elements of water security, as well as its state and interconnections. In addition to national analyses, the framework can also be applied in more limited and specific analyses, such as regional or river basin-level assessments, or more detailed assessments of specific sectors (e.g., agricultural water management and loading, energy production). The analysis suggests that at least the following topics should be priorities in water management and research in Finland: the improvement of ecological status and biodiversity of aquatic ecosystems; understanding of the risks related to emerging harmful substances; adequate investment in water supply networks and agricultural drainage systems; and climate change adaptation and mitigation.

**Author Contributions:** Conceptualisation: all authors; methodology: all authors; software: J.M., M.M., L.A.; validation: all authors; formal analysis: all authors; investigation: all authors; resources: all authors; data curation: J.M., M.M., L.A.; writing—original draft preparation: M.M., S.S., J.M.; —review and editing: all authors; visualisation: all authors; supervision: M.M., M.K.; project administration: M.M., M.K.; funding acquisition: M.K., M.M.

**Funding:** The study was part of the Winland project funded by the Strategic Research Council (303623, 303629). The Winland project looks at Finland's energy, food and water security as well as its resilience through multidisciplinary and interdisciplinary research.

**Acknowledgments:** We thank Antti Belinskij (University of Eastern Finland, SYKE) and Niko Soininen (University of Helsinki) for their assistance with the legislative assessments, and Anni Juvakoski for her assistance with the literature review. We also thank all the interviewees and workshop participants for their important contribution to this study.

**Conflicts of Interest:** The authors declare no conflict of interest. The funders had no role in the design of the study, in the collection, analyses or interpretation of data, in the writing of the manuscript, or in the decision to publish the results.

## Appendix A  Screen Captures from the Excel Software

### Water security assessment - Assessment card for a single criterion

| Crite-rion | 1. State of the water environment<br>1.1 Ecological status of waters (physical-chemical, biology, hydro-morphology) |
|---|---|

| State assessment | **Current state:** | - | Current state is satisfactory or worse than the target level |
|---|---|---|---|
| | **Trend 2030:** | - | State is expected to weaken somewhat by year 2030 |
| | **Reasoning:**<br>The target level of WFD has not been reached in many watercourses: 35 % of the river length, 15 % of lake area and 75 % of the coastal sea area are still in worsta than good condidtion. Climate change is expected to increase the nutrient and solid loads, which consequently increses eutrophication and darkening of the waters, which makes is more difficult to reach the goos state. Of the fish, arctic charr, whitefish and trout require cold water, but pike perch benefits from eutrophication and warming. | | |

| Linkages to food and energy security | **Linkages** | | **Reasoning:** |
|---|---|---|---|
| | **Water → Energy** | ** | Improving the state of important migratory fish requires concessions from the hydro power companies (e.g. large enough streams, more ecological regulation practices). On the other hand, hydro power is an important factor in the energy mixture by providing means to store energy to reservoirs. Peat extraction is not allowed in the areas, where achieving the good state of waters is jeopardized. |
| | **Energy → Water** | ** | Peat extraction and hydro power production has an impact to the biota and migratory fishes of rivers and lakes. Condennsation water of nuclear and fossil power plants cause local warming on watercourses that has an impact to the biota and fishes. Sulphur and nitrogen dioksides cause acidifying fallout to the lakes, although nowadays removal of these particles from the emissions is very efficient. |
| | **Water → Food** | * | Increasing the quality of surface and ground water has positive impact to agriculture (irrigation water) and to the quality of the water used in food production. The quality of the water has an impact to shares of the different fish species and consequently to the amount of fish catches. |
| | **Food → Water** | ** | The loading from farming and fish farming has a negative impact to the ecologial state of the waters. Irrigation and other water intake can affect to the amount of waters in small water courses. Watercoures regulation is ofter carried out on the basis of the needs of farming (e.g. lowering the water levels during spring time). |

| Legislation | **Functionality:** | - | Legislation is partly outdated and there is a need for updating it |
|---|---|---|---|
| | **Reasoning:**<br>The state targets of the water framework directive has not been fully reached. The consideration of permissions and the relationships between state of the waters and controlling them are not clear in the legislation. Means of legislation to reduce scattered loading are restricted. | | |

**Figure A1.** *Cont.*

| Crite-rion | **1. State of the water environment** |
|---|---|
| | **1.1 Ecological status of waters (physical-chemical, biology, hydro-morphology)** |

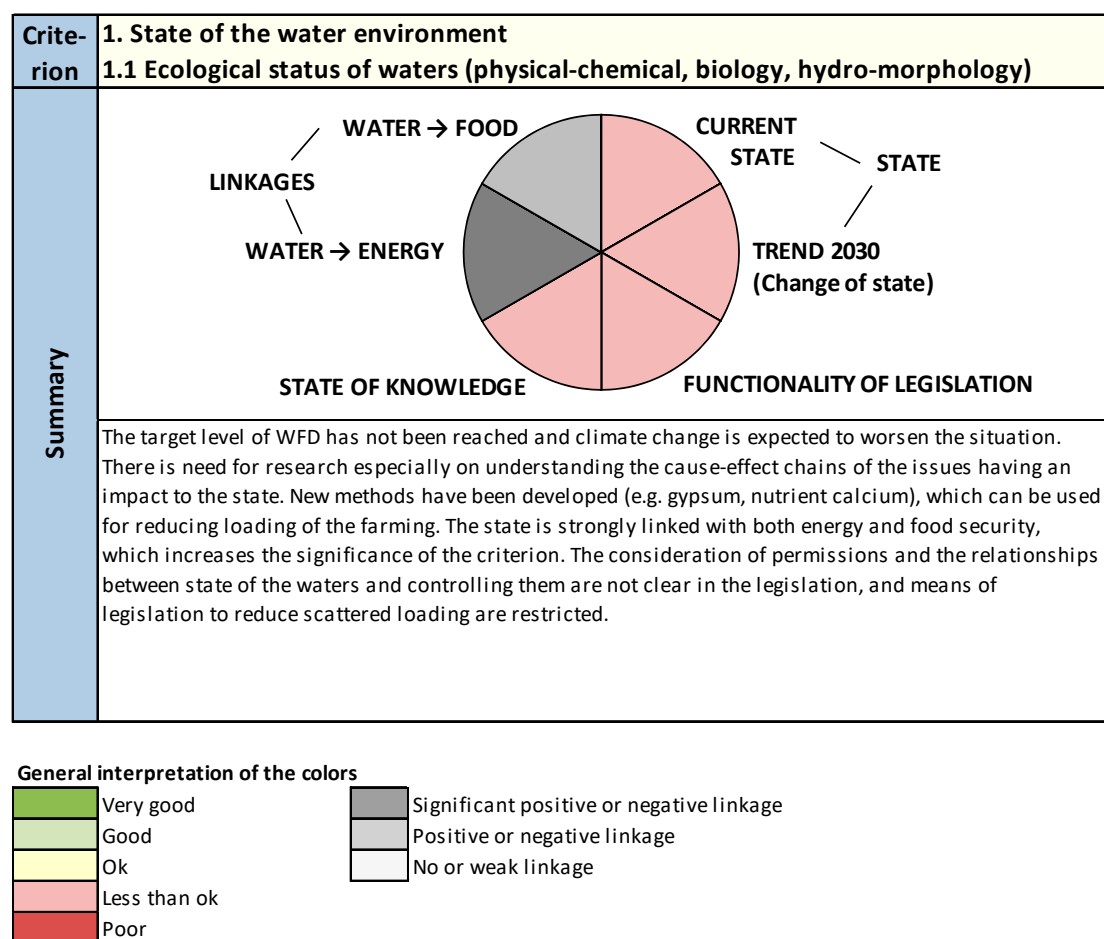

**Summary**

The target level of WFD has not been reached and climate change is expected to worsen the situation. There is need for research especially on understanding the cause-effect chains of the issues having an impact to the state. New methods have been developed (e.g. gypsum, nutrient calcium), which can be used for reducing loading of the farming. The state is strongly linked with both energy and food security, which increases the significance of the criterion. The consideration of permissions and the relationships between state of the waters and controlling them are not clear in the legislation, and means of legislation to reduce scattered loading are restricted.

**General interpretation of the colors**

| | | | |
|---|---|---|---|
| | Very good | | Significant positive or negative linkage |
| | Good | | Positive or negative linkage |
| | Ok | | No or weak linkage |
| | Less than ok | | |
| | Poor | | |

**Figure A1.** Example of an assessment card for a single criterion (ecological status of water).

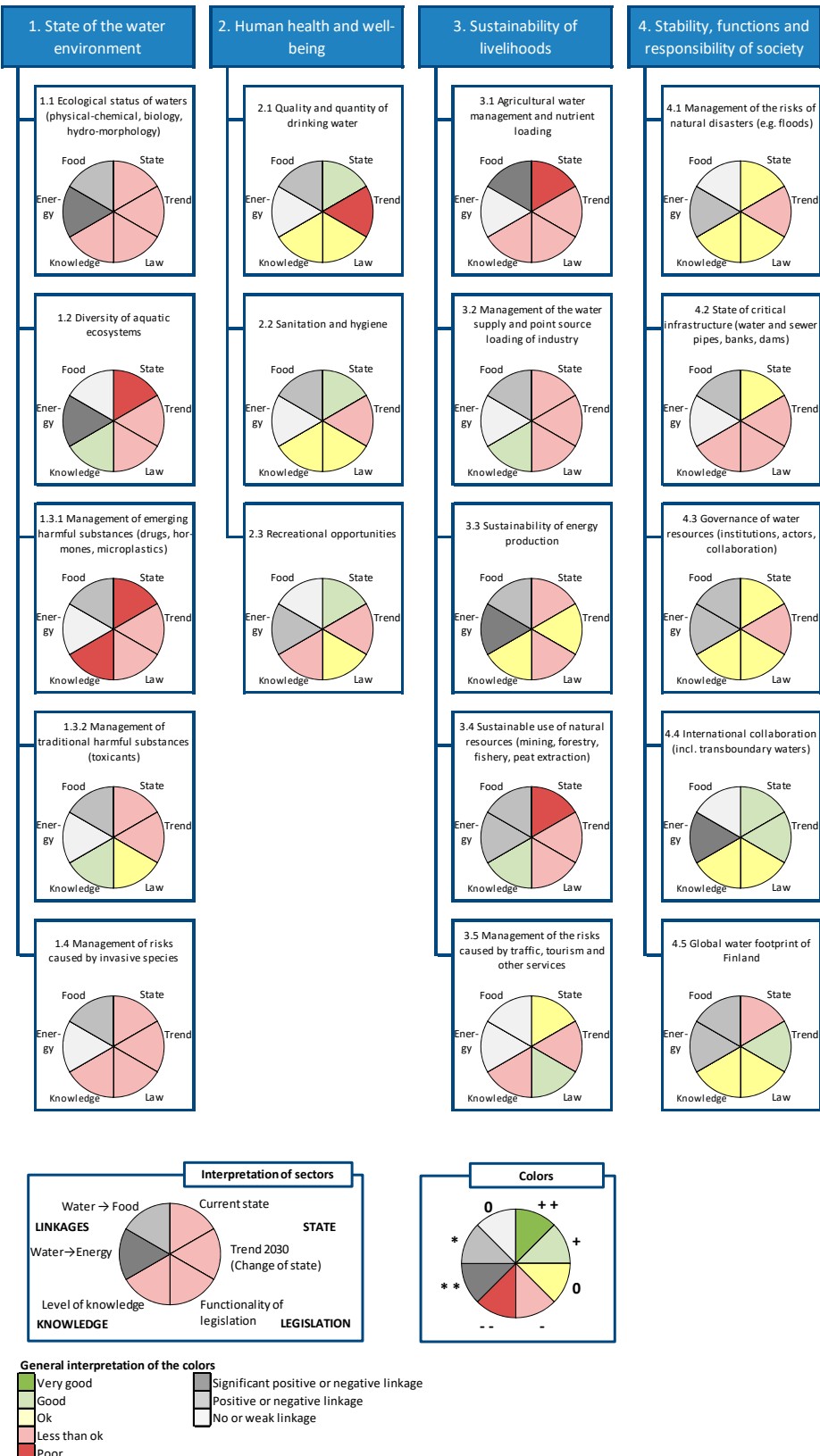

**Figure A2.** Sector charts for the criteria.

## Appendix B  Detailed Information about the Water Security Criteria

**Table A1.** Descriptions of the water security criteria.

| Criteria | Description |
|---|---|
| **1. State of the aquatic environment** | |
| 1.1. Ecological status (physicochemical, biology, hydro-morphology) | Describes the physicochemical, biological and hydro-morphological status of surface water ecosystems using the ecological classification system of Water Framework Directive (WFD). |
| 1.2. Diversity of aquatic ecosystems | This criterion describes diversity and surface water that are not included in the ecological classification system of WFD; e.g., small water bodies (wells and springs), water bodies created by the uplift of land. In addition, endangered species are included, e.g., land-locked salmon, lake and sea trout, Saimaa ringed seal. |
| 1.3. Management of harmful substances in the water courses (toxicants, drugs, hormones, microplastics) | Includes both "old" and "well-known" harmful substances, like heavy metals, DDT, dioxin and new emerging chemicals (e.g., pesticides, pharmaceuticals and personal care products, fragrances, plasticisers, hormones, flame retardants, nanoparticles, perfluoroalkyl compounds, chlorinated paraffins) and plastic pollution. Both surface and groundwater are included; 33 priority substances that are known to be harmful or dangerous at EU level and included in the ecological classification are not considered here to avoid overlap. |
| 1.4. Management of risks caused by invasive species | An invasive species is a species that is not native to a specific location (an introduced species), and that has a tendency to spread to a degree believed to cause damage to the environment, human economy or human health. |
| **2. Human health and well-being** | |
| 2.1. Quality and quantity of drinking water | The quality and quantity of household water is estimated on the basis of average water quality and availability. In addition, the assessment takes into account individual events (water crises) that weaken quality/quantity and their frequency. |
| 2.2. Sanitation | Sanitation is assessed on the basis of how well the treatment of municipal waste water is handled on average. In addition, the assessment takes into account the possible individual events (water crises) that reduce wastewater treatment and their frequency. |
| 2.3. Recreational opportunities | Recreational values are evaluated on the basis of the value added by the aquatic environment for recreation. Includes swimming and fishing opportunities, but also the landscape and cultural values of the aquatic environment. Increase in water turbidity, disadvantages caused by eutrophication (e.g., massive/toxic algae blooms) and decrease in the public access to water bodies (e.g., construction of shoreline) diminish recreational opportunities. |
| **3. Sustainability of livelihoods and industry** | |
| 3.1. Agricultural water management and nutrient loading | Good conditions of the fields, soil (e.g., amount of humus), drainage and irrigation systems, influence considerably on the crop and nutrient loading to the water bodies. In addition, the use of fertilisers, status of water protection measures in fields and farms are assessed (e.g., protection zones, wetlands, two-stage channels, sludge treatment). |
| 3.2. Water supply and point-source loading of industry | Quantity and quality of water used by industry and treatment of wastewater discharges from the plants. Includes, e.g., pulp and paper industry, chemical industry, ore enrichment plants. |
| 3.3. Energy production (hydro power, nuclear, peat extraction, wind power) | The assessment includes, in particular, the sustainability of hydropower and peat extraction (peat is an important source of energy in Finland; peat extraction area has varied annually between ca. 40,000 and 60,000 ha) but also impacts of cooling water of nuclear power plants as well as other power plants. |
| 3.4. Sustainable use of natural resources (forestry, mining, fishery) | The management, production and harvesting of natural resources, such as forests (felling, drainage of forest areas), aquaculture and ore extraction. |
| 3.5. Management of the risks caused by traffic, tourism and other services | Includes transport (e.g., oil and chemical accidents in roads, railways and waterways), tourism and sales services (gas stations, etc.). For example, the accidents of oil tankers particularly in the Gulf of Finland and chemical accidents in land transport can cause significant impacts on aquatic ecosystems. |
| **4. Stability, functions and responsibility of society** | |
| 4.1. Reduction of the natural disaster risks (floods, droughts) | Prevention, preparedness and response of natural risks; in particular flooding and drought. |
| 4.2. State of critical infrastructure (water and sewer pipes, banks, dams) | Status of critical infrastructure as defined in the Social Security Strategy (2017), incl. water supply infrastructure, banks and dams. |
| 4.3. Governance of water resources (institutions, actors, collaboration) | Water resources management covers organisations and stakeholders responsible for water as well as institutional frameworks that regulate the interaction between them (policy, strategies, laws). |
| 4.4. International collaboration (including transboundary water management) | International cooperation in the water sector, incl. transboundary cooperation. It can be assessed in terms of its magnitude (share of boundary waters with existing agreements) or with the criteria of UNECE, Global Water Partnership, World Bank or Strategic Foresight Group. |
| 4.5. Global water footprint of Finland | Impact of Finnish consumption, production and investment on water resources and water safety outside Finland. Measurements include: water footprint (see, e.g., https://wwf.fi/mediabank/2306.pdf), water risks (e.g., http://waterriskfilter.panda.org/) and commitment to water responsibility (see, for example, water liability https://commitment2050.com/browse-#commitments/details/59254488D4DF3C0D1C6027FA). |

**Table A2.** Examples of research information and indicators used in the water security assessment.

| Water Security Criteria | Examples of Indicators | Source |
|---|---|---|
| **1. State of the water environment** | | |
| 1.1. Ecological status of water | Ecological status of surface water | https://www.ymparisto.fi/en-US/Waters/State_of_the_surface_waters |
| 1.2. Diversity of aquatic ecosystems | Threatened inland water species | https://www.biodiversity.fi/en/habitats/inland-waters/iw11-threatened-inland-water-species |
| 1.3.1. Management of emerging harmful substances | Microplastics and drugs in wastewater | Talvitie, J. et al. 2015[1] Kankaanpää, A. et al. 2014[2]. |
| 1.3.2. Management of traditional harmful substances | Loading of heavy metals from industry | https://www.biodiversity.fi/en/habitats/inland-waters/iw3-harmful-substances |
| 1.4. Management of risks caused by invasive species | Alien inland species | https://www.biodiversity.fi/en/habitats/invasive-species/as2-alien-inland-water-species |
| **2. Human health and well-being** | | |
| 2.1. Quality and quantity of drinking water | SDG 6.1.1 Proportion of population using safely managed drinking water services | http://pxnet2.stat.fi/PXWeb/pxweb/en/SDG/ Gunnarsdottir, M. J. at al. 2017.[3] |
| 2.2. Sanitation and hygiene | SDG 6.3.1 Proportion of wastewater safely treated | http://pxnet2.stat.fi/PXWeb/pxweb/en/SDG/ |
| 2.3. Recreational opportunities | Proportion of bathing water sites with excellent water quality | http://ec.europa.eu/environment/water/water-bathing/index_en.html |
| **3. Sustainability of livelihoods** | | |
| 3.1. Agricultural water management and nutrient loading | Phosphorus load into inland water | https://www.biodiversity.fi/en/habitats/inland-waters/iw1-phosphorus |
| 3.2. Management of the water supply and point source loading of industry | Nutrient discharges into surface water | https://www.ymparisto.fi/en-US/Maps_and_statistics/The_state_of_the_environment_indicators/Fresh_water_and_the_sea/Nutrient_discharges_from_industry_and_co%2828956%29 |
| 3.3. Sustainability of energy production | Coverage of regulation development projects | https://www.biodiversity.fi/en/habitats/inland-waters/iw15-regulation-development |
| 3.4. Sustainable use of natural resources | Area used for peat production | https://www.biodiversity.fi/en/habitats/mires/mi3-peat-production |
| 3.5. Management of the risks caused by traffic, tourism and other services | Maritime transport | https://www.biodiversity.fi/en/habitats/baltic-sea/bs4-maritime-transport |
| **4. Stability, functions and responsibility of society** | | |
| 4.1. Management of the risks of natural disasters | Flood damages and flood risk management | https://www.ymparisto.fi/en-US/Waters/Floods/Flood_risk_management/Flood_risk_management_planning |
| 4.2. State of the critical infrastructure | SDG 6.a.1 Amount of water- and sanitation-related official development assistance, euros | http://pxnet2.stat.fi/PXWeb/pxweb/en/SDG/ |
| 4.3. Governance of water resources | General governance indicator | https://info.worldbank.org/governance/wgi |
| 4.4. International collaboration | SDG 6.5.1 Degree of integrated water resources management implementation; SDG 6.5.2 Proportion of transboundary basin area with an operational arrangement for water cooperation, Water cooperation quotient | https://www.strategicforesight.com/publication_pdf/Water%20Cooperation%20Quotient%202017.pdf |
| 4.5. Global water footprint of Finland | Sustainability of water footprint | https://waterfootprint.org/en/resources/waterstat/national-water-footprint-statistics/ https://waterfootprint.org/en/resources/waterstat/water-pollution-level-statistics/ https://waterfootprint.org/en/resources/waterstat/water-scarcity-statistics/ |

[1] Talvitie, J. et al. 2015. Do wastewater treatment plants act as a potential point source of microplastics? Preliminary study in the coastal Gulf of Finland, Baltic Sea. Water Science and Technology: a journal of the International Association on Water Pollution Research. [2] Kankaanpää, A. et al. 2014. Use of illicit stimulant drugs in Finland: a wastewater study in ten major cities. Science of the Total Environment, 487, 696-702.72. 1495-1504. 10.2166/wst.2015.360. [3] Gunnarsdottir, M. J. at al. 2017. Status of small water supplies in the Nordic countries: Characteristics, water quality and challenges. International journal of hygiene and environmental health, 220(8), 1309-1317.

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
