# Peer review of "A Framework for Assessing Water Security and the Water–Energy–Food Nexus—The Case of Finland"

_sustainability, doi:10.3390/su11102900_

Round 1
Reviewer 1 Report
Congratulations on an interesting case of stakeholder engagement with experts that addresses an important topic comprehensively. You may have oversold the project as unique, (See for example, "Explore, Synthesize, and Repeat: Unraveling Complex Water Management Issues through the Stakeholder Engagement Wheel", Kelly E. Mott Lacroix * and Sharon B. Megdal, in Water.) but it explores the challenges and choices in a way that adds to the existing literature.
Some terms could use definition or clarification where they are first used: functionality of legislation, functionality and responsibility of the society, risks caused by traffic (transportation of potential contaminants?), target levels, significant impacts, loading. In the discussion lines 290-319 and 383 to 394, the text should make it clear that the relationships are either positive or negative; that is, in a positive relationship a positive movement in a water criterion leads to a positive movement in the food/energy criterion or a positive movement in a water criterion leads to a negative movement in the food/energy criterion, etc. This would also clarify what is meant by "or alternatively it can be in contradiction to that." Then at line 387, change "impact" to "relationship".
Graphics - Table 2 was too small to read. I suggest inserting inserting an example of the visualizations at line 343.
The draft could use a review for minor language errors. On line 104, for example, I believe you mean "exception" rather than "exemption" and on line 419 - "side" instead of "shade".. There are also many small problems with the use of articles (a/an, the) and prepositions.
Author Response
REVIEWER 1
Thank you very much for the valuable comments which helped us to improve considerably the paper.
Congratulations on an interesting case of stakeholder engagement with experts that addresses an important topic comprehensively. You may have oversold the project as unique, (See for example, "Explore, Synthesize, and Repeat: Unraveling Complex Water Management Issues through the Stakeholder Engagement Wheel", Kelly E. Mott Lacroix * and Sharon B. Megdal, in Water.) but it explores the challenges and choices in a way that adds to the existing literature.
Response: Thank you very much for the positive feedback and for the interesting article. We agree that in water management field collaborative approaches have been applied for a long time and in this respect our study is not unique. However, in water security assessment, most of the papers rely on the statistical indexes which have been developed by water management expert.
Some terms could use definition or clarification where they are first used: functionality of legislation, functionality and responsibility of the society, risks caused by traffic (transportation of potential contaminants?), target levels, significant impacts, loading.
Response: Thank you very much for this good comment. We have clarified the terminology in the paper as follows:
Environmental risks of traffic: “includes e.g. oil and chemical accidents in roads, railways and waterways” (lines 211-212)
Functionality of legislation: “i.e. how up to date the legislation is and how flexible it is for changing circumstances” (lines 225-226)
Target levels: “With regard to some criteria a precise target level exists. For Ecological status of waters, target level is good ecological status according to the Water Framework Directive. For some other criteria (e.g. Sustainability of energy production), target levels were more vague and the assessment was based on the experts' views about the prevailing situation compared to an ideal situation.” (Footnote1 p. 7)
Significant impact: “i.e. a noticeably or measurably large amount” (lines 318-319)
Loading: we have clarified what we mean with loading in text, tables and figures:.
3.1 Agricultural water management and nutrient loading
3.2 Management of the water supply and point source loading of the industry
In the discussion lines 290-319 and 383 to 394, the text should make it clear that the relationships are either positive or negative; that is, in a positive relationship a positive movement in a water criterion leads to a positive movement in the food/energy criterion or a positive movement in a water criterion leads to a negative movement in the food/energy criterion, etc. This would also clarify what is meant by "or alternatively it can be in contradiction to that." Then at line 387, change "impact" to "relationship".
Response: Thank you for this comment. We have added following text to clarify that both positive and negative relationships can occur. “In practice, the impact can either be positive or negative. A positive relationship means that the positive movement of a criterion in a row leads to a positive movement of a criterion in a column, and correspondingly, a negative movement in a row leads to a negative movement in a column. In a negative relationship, the impacts are the other way around.” (lines 349-353).
Graphics - Table 2 was too small to read. I suggest inserting inserting an example of the visualizations at line 343.
Response: We have enlarged Table 2 and added an example to the text.
“For example, improvement in the Ecological status of waters improves the Diversity of aquatic ecosystems as the living conditions for salmonids that prefer oligotrophic conditions improve.” (lines 353-355)
The draft could use a review for minor language errors. On line 104, for example, I believe you mean "exception" rather than "exemption" and on line 419 - "side" instead of "shade".. There are also many small problems with the use of articles (a/an, the) and prepositions.
Response: The language has now been proofread by a professional and a large number of prepositions and articles, for instance, have been corrected. We have also corrected the language on lines 104 and 419. In addition, we have clarified / polished text in many places.

Reviewer 2 Report
Dear authors,
Please read the attached document. It has my suggestions towards improving the article.

Author Response
REVIEWER 2
Thank you very much for the very good comments regarding e.g. the structure of the article.
Line 86-90 I recommend replacing the phrase in Section 2 with Part 2 or Chapter 2
Response: Section is commonly used in scientific articles and chapters in books. We also investigated two earlier papers published in Sustainability and in both of them section was used.
Line 151 There’s an error in text and citations, double-check references.
Line 197 You introduced the error into the text and made references
Line 200 Figure 2 is not quoted in the text, please do so.
Line 288 is quoted in table 2, but, regarding Table 1: where it was quoted and presented?
Table 1 is not quoted in the text, please do so before submitting it.
Line 372 I reccommend introducing the text into a quoted, corrected and verified letter and references.
Response: Thank you for these observations. Unfortunately, we did not notice before the submission of the paper that the some cross-reference links in the Word file had broken between references to some tables and figures, and all these errors relate to that. They have been corrected.
Line 193 There is "see Appendix B" and in line 208 "see Appendix A" - What is the logic of the order of presentation?
Response: Thank you for this good observation. We have changed the order of Appendixes.
Don’t you understand that you started with a Table 2 discussion, can you present Table 1?! Table 2 is positioned on page 11 three pages away from where it was quoted in the text (page 8). It's hard to follow and track the text of the article.
Response: Thank you very much for this valuable comment. We agree with the reviewer that distinguishing method description and results made it difficult to follow the text. Therefore, we have moved text from Subsections 3.3.1-3.3.3 to Section 4 where the results of the assessment are presented. In our view, these changes improved the readability of the paper.
Table 2 is hard to read, change the size of the characters.
Response: We have enlarged Table 2
General Recommendation: In my opinion, the article is a little too extensive and could be more restricted to easily navigate the article. An integration into the text and materials presented in Appendices A and B help to increase the value of an article and to present the logic of a methodology for using the information processing methodology.
Response: Thank you very much for the recommendation. We have tried to shorten the text where possible without losing relevant information. We have, for example, removed text which described the comments and feedback we received from the experts during the development process (lines 223-239 in the first submission). We have also moved Figure B.2 and B.5 from the Appendix B into the text and removed Figure B.1 .

Round 2
Reviewer 2 Report
The article looks much better in this form.